# Impact of *Spirulina maxima* Intake and Exercise (SIE) on Metabolic and Fitness Parameters in Sedentary Older Adults with Excessive Body Mass: Study Protocol of a Randomized Controlled Trial

**DOI:** 10.3390/ijerph18041605

**Published:** 2021-02-08

**Authors:** Marco Antonio Hernández-Lepe, José de Jesús Manríquez-Torres, Omar Ramos-Lopez, Aracely Serrano-Medina, Melinna Ortiz-Ortiz, Jorge Alberto Aburto-Corona, María del Pilar Pozos-Parra, Luis Eduardo Villalobos-Gallegos, Genaro Rodríguez-Uribe, Luis Mario Gómez-Miranda

**Affiliations:** 1Facultad de Medicina y Psicología, Universidad Autónoma de Baja California, Tijuana 22390, Mexico; marco.antonio.hernandez.lepe@uabc.edu.mx (M.A.H.-L.); jose.de.jesus.manriquez.torres@uabc.edu.mx (J.d.J.M.-T.); oscar.omar.ramos.lopez@uabc.edu.mx (O.R.-L.); serrano.aracely@uabc.edu.mx (A.S.-M.); maria.pozos@uabc.edu.mx (M.d.P.P.-P.); villalobos.luis@uabc.edu.mx (L.E.V.-G.); genaro.rodriguez@uabc.edu.mx (G.R.-U.); 2Facultad de Deportes, Universidad Autónoma de Baja California, Tijuana 22390, Mexico; melinna.ortiz@uabc.edu.mx (M.O.-O.); jorge.aburto@uabc.edu.mx (J.A.A.-C.)

**Keywords:** *Arthrospira maxima*, physical exercise, randomized trial, obesity, elderly, dyslipidemia

## Abstract

Life expectancy has increased unprecedentedly in recent decades, benefiting the longevity of the world’s population. The most frequent pathological conditions presented in this age group include excessive body fat, frailty, and hypercholesterolemia. These pathological characteristics condition general health and autonomy in adults to carry out their usual activities. In this sense, the search for a healthy lifestyle is necessary, consisting in a healthy diet that includes supplementation with nutraceuticals and the daily practice of physical activity. This study protocol aims to evaluate the independent and synergistic effect of 12 weeks of *Spirulina maxima* intake (5 g/day), with or without an exercise program on metabolic and fitness parameters of 52 sedentary older adults with excessive body mass in a double-blind, randomized, crossover, controlled trial design. The main findings from this trial will provide novel evidence for future interventions designed for the elderly population and the result will be disseminated through peer-reviewed journals and international meetings. ClinicalTrials.gov identification number: NCT04658875 (Effect of *Spirulina maxima* and Exercise on General Fitness and Blood Lipids in Older Adults).

## 1. Introduction

Life expectancy has increased unprecedentedly in recent decades, benefiting the longevity of the World’s population, with projections of a growth in the population older than 60 years from 900 million in 2015 to two billion in 2050 [1], this means that the older population will move from 12% to 22% of the global population in the next years. For the elderly population, the most frequent health conditions include excessive body fat, frailty, and hypercholesterolemia [2]. Some of these conditions can co-exist in aging individuals, presenting new challenges, for example, the combination of excessive body fat and reduced muscle mass or strength, recently defined as sarcopenic obesity [3]. Muscle protein breakdown is one of the main contributors to chronic low-grade inflammation and sarcopenia. Recent studies suggest that the progression of muscle fiber anabolism and muscle wasting is highly related to the reduction of the endogen antioxidant system and oxidative stress, inducing the activation of muscular autophagy mechanisms [4].

These phenomena are a major concern, given that it compromises general health, physical ability (i.e., autonomy to carry out their daily life activities such as walking, eating, preparing or buying food, and bathing, among others) and mortality rate [5]. That means that the elderly population are at increased risk for developing some disease or dependence during their last years of life.

Overweight, obesity and dyslipidemias contribute considerably to the burden of cardiovascular diseases and exacerbate the age-related decline in physical function, which causes frailty and impairs quality of life [6]. Many of the available conventional treatments aimed at reducing these risk factors often have collateral side effects, leading older people to often use complementary medicine instead. Thus, the pharmacological industry has focused on developing new treatment options, in this sense, the search for a healthy lifestyle is needed, which covers a healthy diet that includes supplementation with nutraceuticals and daily physical activity [7].

Regarding supplementation with nutraceuticals, the cyanobacterium *Spirulina maxima* (*S. maxima*) presents antioxidant compounds, which has been associated with beneficial cardiovascular effects [8] also it has important benefits to reduce non-communicable diseases, alone or in combination with exercise, mainly in animal models [9]. Recently, studies on *S. maxima* effects focus on verifying its components’ biological activity, including general fitness improvements and hypolipidemic effects [10]. It has been shown that liver lipid accumulation (as well as the visceral in general), can be reduced by modulating the infiltration of macrophages by Spirulina compounds [11]. However, given that these investigations were mainly carried out in animals, appropriate clinical trials are necessary [12] to add knowledge in the beneficial effect of *S. maxima* in humans.

Furthermore, performing exercise has been widely reported to reduce the risk of hypertension, diabetes, and hypercholesterolemia [13]. In this context, moderate intensity-physical exercise has exerted the best protective effect, mainly attributed to metabolic adaptations [14]. The effects of performing exercise programs in older adults have shown a reduction on adiposity and serum lipid levels [15], and also a reduction in sarcopenia markers, such as improvement in muscle mass, muscle strength, and physical performance [16]. Exercise has also been included as part of multicomponent interventions targeting physical function in older adults living in poverty in rural communities with significant results [17]. Thus, appropriate exercise prescription methods need to be implemented, in this sense, the rating of perceived exertion scale [18] and the Talk Test [19] are no-invasive, easy access, and validated subjective methods to estimate exercise training intensity by self-evaluation of the intensity of a physical activity perceived and the difficulty to speak comfortable during a 30 speech, respectively.

Therefore, we hypothesize that the administration of *S. maxima* alone or together with an exercise plan will decrease the body fat percentage and blood lipids while increasing physical function in sedentary elderly patients with excessive body mass in a randomized, double-blind, counterbalanced, crossover, controlled trial. The main objective of our research is to evaluate the independent and synergistic effect of 12 weeks of *S. maxima* intake (compared to placebo), with or without an exercise program on the body composition (body fat percentage, muscle mass percentage, and body mass index (BMI)) in sedentary older adults with excessive body mass. The secondary objective is to test the effects of both interventions in the blood lipid profile, energy intake, physical function, sleep wake patterns, mental health, and inflammation, as well as to analyze the influence of the genetic background in all outcomes in older adults with excessive body mass.

## 2. Materials and Methods

This study is designed according to the standard protocol requirements for clinical trials (SPIRIT) (Appendix A) and will be conducted and reported in keeping with the Consolidation Standards of Reporting Trials (CONSORT).

### 2.1. Eligibility Criteria

Fifty-two sedentary (daily energy expenditure <4 METs, as measured by a continuous physical activity questionnaire (IPAQ)) [20] older adults (≥60 years) with a BMI over 25 kg/m^2^ volunteered to participate will be recruited by personalized interviews including a Physical Activity Readiness-Questionnaire for Everyone (PAR-Q+) [21] conducted to ensure eligibility. The participants inclusion and exclusion/elimination criteria are depicted in Table 1.

### 2.2. Interventions

The clinical trial consists of two interventions: (1) supplementation with *S. maxima* (5 g/day) or placebo (5 g/day of a low-calorie saccharine), and (2) an exercise program, during a period of 24 weeks in a randomized, double-blind, counterbalanced, crossover, controlled trial with a 2 × 2 factorial design. Presentation of both supplements will be powder encapsulated in dark capsules of 0.5 g to present the same organoleptic attributes and mask their individual characteristics.

Inclusion in the exercise program intervention will be decided by each participant, who will be randomly assigned to one of the two possible nutritional interventions (*S. maxima* or placebo) categorized into four treatments: an exercise program plus *S. maxima* intake; exercise with placebo administration; *S. maxima* intake without exercise, or placebo supplementation without exercise program (Control). To assess compliance to the supplement intake, a crossover will be conducted for the supplementation interventions.

The first period of treatment will be carried out for 12 weeks, followed by a four-week wash-out period to avoid any possible delayed effect of *S. maxima* in the organism. Finally, a further 12 weeks of treatment for the second intervention groups (Figure 1).

Due to absence of data related to wash-out and treatment periods length, we considered 4 and 12 weeks, respectively, as stated by a systematic review of clinical trials that used spirulina as treatment [22].

#### 2.2.1. Exercise Program

Participants in the exercise program groups will exercise five days a week. The physical activity program will be conducted in accordance with the American College of Sports Medicine recommendations [23] and will consist of Monday to Friday chair exercises performed during 24 weeks.

The approximate exercise duration will be of 40–50 min. Each exercise session has an initial or warm-up phase of 5–10 min (joint movement exercises to increase body temperature starting with movements of head, upper extremities trunk, hip, and, lower extremities); a main physical conditioning phase of 25–30 min that will consist of developing a personalized circuit of strength exercises of the main muscle groups (arms, shoulder, chest, back, and legs) at moderate intensity (between 3.0 and 6.0 metabolic equivalents of task (METs)) using rating of perceived exertion scale and the Talk Test; finally, a cool-down or relaxation phase of 5–10 min. To define the individual exercise intensities in the main physical conditioning phase, participants will have a training of 3 to 5 days before the beginning of the clinical trial to learn how to use the rating of perceived exertion scale and the Talk Test as markers of exercise training intensity.

Trained personnel will assist older adults to perform the exercise program, including articular flexibility, postural exercises, and proprioception exercises. These activities will be selected according to the individual deterioration and joint stiffness that the participants could present. Researchers are trained in cardiopulmonary resuscitation and the laboratory has the necessary tools and communication channels to carry out emergency procedures if a participant presents any side effect of exercise.

#### 2.2.2. Adherence Assessment

Participants will visit the laboratory weekly to receive new supplements and related information, which will be registered on case report formats. Treatment adherence will be assessed by counting the remaining capsules when the participants return to the laboratory. Participants will be asked to report as soon as an adverse effect occurs to analyze the possibility of suspending supplementation.

### 2.3. Outcome Measures

Primary outcome will be the change in body composition (body fat percentage, muscle mass percentage, and BMI) of sedentary overweight older adults. We will include as secondary endpoints the energy intake (total calories, proteins, carbohydrates, fat, and micronutrients consumption), genetic markers, and the changes in blood lipid profile (TC, HDL-C, LDL-C, and TG), physical function (grip strength and functional physical condition), sleep-wake patterns, mental health (anxiety, depression, and cognitive performance), inflammatory markers (IL-6 and TNF-α) of sedentary elderly with excess body mass. Health organizations around the world recommend studies focused on improving these response variables [24]. An overview of all variables included in the SIE trial is shown in Table 2.

#### 2.3.1. Body Composition

Body mass will be measured in participants with light clothes and barefoot using an electronic balance and standing height with a clinical stadiometer. Body composition (body fat percentage, muscle mass percentage and BMI) will be measured by three techniques: air displacement plethysmography (BodPod, COSMED, Rome, Italy); electric bioimpedance (InBody770, BIOSPACE, Tokyo, Japan); and anthropometry following the standardized protocols of the International Society for the Advancement of Kinanthropometry (ISAK) [25]. The anthropometric variables to include are the ones of the restricted profile (height, body mass, skinfolds (tricipital, subscapular, bicipital, suprailiac, supraspinal, abdominal, anterior thigh, and medial calf)), girths (relaxed arm, flexed arm, waist, hip, and calf) and skeletal breadths (humerus and femur). The measurements’ margin of error will be <2% for skinfolds and <1% for girths and skeletal breadths. Anthropometric dimensions will be measured twice (thrice when measurements are greater than the margin of error) and recorded as a mean. Researchers are certified as ISAK anthropometrists (Appendix A) and are experienced in handling equipment and procedures. Phase angle (50 kHz) will be obtained from the electric bioimpedance analysis.

#### 2.3.2. Dietary Analysis

Energy intake will be assessed weekly by validated 24-h recalls and a semi-quantitative food frequency questionnaire [26], which allows the quantification of the types and quantities of foods and beverages consumed during the period of interest in the past. Dietary records will be reviewed for missing data by an independent nutritionist and then analyzed for calculation of total calories, proteins, carbohydrates, fat, and micronutrients intake (Diet Analysis Plus, ESHA Research, Salem, OR, USA).

#### 2.3.3. Physical Function

Hand grip strength will be evaluated with a Smedley III handgrip dynamometer (Takei Scientific Instruments Co, Tokyo, Japan), while the seated participant held the dynamometer with the arm on the table, the maximum grip strength of each hand will be obtained twice [27]. A functional fitness test (Senior Fitness Test) will be used to evaluate the functional physical condition [28]. The lower-body strength will be assessed by the *30-s chair stand test*, consisting in getting up and sitting down from a chair during 30 s with arms folded across chest. The upper-body strength will be evaluated by the *30-s arm curl test*, consisting in the number of biceps curls finished in 30 s holding a dumbbell (5 lb for women, 8 lb for men). Lower-body flexibility will be assessed by the *chair sit-and-reach test*, where participants begin in a sitting position with leg extended, then slowly flex the hip joint, reaching the fingers as much as to the tip of the toe. *Back scratch test* will assess upper-body flexibility, where the participants will have one hand reaching over the shoulder and the other one un the middle of the back, the distance between the tips of the extended middle fingers will be assessed independently of the back’s alignment. Agility and dynamic balance will be assessed by the *8-foot up-and-go test*, where participants had to get up of from a chair, walk as fast as possible 2.4 m, turn, and sit down again. Aerobic endurance will be evaluated by the *6-min walk test*, where participants have to walk during 6 min around a 50-yard course.

#### 2.3.4. Blood Lipid Profile

Refrigerated blood samples will be centrifugated at 3000 g for 20 min to obtain the plasma, which will be processed to determine total cholesterol (TC), high-density lipoproteins associated cholesterol (HDL-C), low-density lipoproteins associated cholesterol (LDL-C), and triglycerides (TG) by using standard colorimetric procedures (Spinreact, Girona, Spain) with an auto-biochemistry analyzer Mindray BS200 (MINDRAY, Shenzhen, China).

#### 2.3.5. Mental Health and Cognitive Performance

The Montreal Cognitive Assessment (MoCA) will be used to evaluate cognitive performance [29]. Anxiety symptoms will be measured using the Generalized Anxiety Disorder-7 (GAD-7) questionnaire [30]. To assess depressive symptoms, the Patient Health Questionnaire-9 (PHQ-9) will be used [31]. These instruments will be applied in the baseline evaluation and in each of the weekly visits.

#### 2.3.6. Sleep-Wake Patterns

Recording of daily activity and sleep-wake patterns of the participants will be made from actigraphy of continuous recording [32], with three-dimensional accelerometer technology (Mini Mitter, Philips Respironics Inc., Bend, OR, USA) to increase the sensitivity of the instrument. The monitoring will begin one day before starting the study to validate standard sleep patterns and will end once the last measurement is finished, the study will be complemented with the Pittsburgh Sleep Quality Questionnaire [33].

#### 2.3.7. Inflammatory Markers

Plasma concentrations of IL-6 and TNF-α will be measured by ELISA kits using an automated analyzer system (Triturus, Grifols, Barcelona, Spain). IL-6 will be evaluated by means of Quantikine immunoassay kit from R&D Systems (Minneapolis, MN, USA). Serum levels of TNF-α will be evaluated using high sensitivity commercial ELISA with an alkaline phosphatase signal amplification system (Quantikine HS, high sensitivity, R&D Systems, Minneapolis, MN, USA).

#### 2.3.8. Genetic Analyses

With the purpose of evaluate the influence of the genetic make-up in adiposity and metabolic changes after interventions, two common single nucleotide polymorphisms related to obesity (rs9939609, *FTO*) and lipid metabolism (rs1799883, *FABP2*) will be detected by Taqman probes (Applied Biosystems, Foster City, CA, USA) through a RT-PCR system.

### 2.4. Allocation and Sample Size

The initial allocation will be performed in such a way that each group had almost the same proportion of *S. maxima*:placebo (1:1) supplemented individuals through the RandomizeR package of the R version 4.0.2 statistical software [34]. Participants’ group allocations will be performed by an independent researcher, who does not have any other involvement during the study, break of double-blind (participants and principal investigators) can occur only in exceptional circumstances and must be immediately reported. Sample size was calculated considering a level of significance of 0.05 with a study power of 0.85, selecting a sample of ≥52 participants. The statistical software G*Power (v3.1.9.7, Universität Kiel, Düsseldorf, Germany) [35] was used to determine the sample size.

### 2.5. Data Collection

Eligible participants will visit the body composition laboratory at UABC in Tijuana, Mexico, between 7–9 AM. They will assist during 3–5 days before the onset of the trial for pre-intervention tests (electrocardiogram, blood chemistry, and a check-up performed by a certified doctor) and familiarization with the exercise program. The first session will be the beginning of the intervention (Day 0), here, antecubital blood samples will be drawn by an expert clinician into EDTA tubes after overnight fasting, after that, body composition measures will be performed, finally, dietary, psychological, and physical activity questionnaries will be obtained. All procedures will be repeated on days 84, 112, and 196. Standard room temperature and humidity will be checked constantly.

### 2.6. Statistical Methods

Normality and homoscedasticity of variables will be screened by the Shapiro–Wilk and the Levene tests, for each group. The analysis of statistical differences among treatments and time, univariate repeated measures ANOVA with two within-participants (initial and final values), and four inter-participants (treatments) factors will be used. Moreover, basal and post-treatment values will be contrasted by paired *t*-tests. Delta values will be analyzed by one-way ANOVA and Tukey Post-hoc tests (normal distribution and equal variances) and Kruskal-Wallis H and Dunn’s Post-hoc test (non-parametric distribution and unequal variances), with Bonferroni setting for multiple comparisons. Spearman correlations will be run to evaluate the associations among variables. Eventual gen-diet and gen-exercise interactions concerning anthropometric and metabolic outcomes will be assessed by univariate regression analyses. Statistically significance will be set at a *p*-value of 0.05. All statistical analyses will be completed using IBM SPSS package version 23 (SPSS Inc, IBM Company, Chicago, IL, USA) and STATA software (StataCorp, College Station, TX, USA).

### 2.7. Ethics

Participants will be informed of the study rationale as well as all procedures. Their inclusion will be formalized by signing an informed consent (Appendix A), being anonymity and confidentiality carefully enforced, and all participants data will be stored during five years after the study completion. The protocol and procedures have been approved by the Autonomous University of Baja California (UABC) bioethics committee (Appendix A). One UABC scientific committee member and one UABC bioethics committee member will have always access to the interim results and can decide to terminate the trial according the adverse events or unintended effects that could be presented during trial conduct. The clinical trial has been registered prospectively at ClinicalTrials.gov database (Date of approval: 10 December 2020; Identification number: NCT04658875), moreover, the World Medical Association’s Declaration of Helsinki guidelines [36] will be followed.

## 3. Discussion

The SIE study will be the first examining the synergistic effect of *S. maxima* intake and exercise on adiposity, physical function, serum lipids, mental health, sleep-wake patterns and inflammation, as well as to analyze the influence of the genetic background in these outcomes in older adults, using a double-blind, randomized, crossover trial. Because of the double-blind design, the described methodology and a higher number of participants than other studies [22], it is expected to achieve better support for the hypothesis that *S. maxima* will have hypolipidemic effects and improve on adiposity and physical function together with an exercise program in sedentary older adults with excessive body mass. It is noteworthy that not every overweight or obese patient has problems with lipid metabolism. The association between increase of body fat and lipid alterations is well known [37]. For that reason, the primary focus of this protocol is prevention, which is why we do not consider dyslipidemia as an inclusion criterion even though it could be present in this condition. Finally, it is known that the practice of exercise and a balanced diet stimulate the reduction of body mass in obese people and generally decreases cardiometabolic risk factors, so it probably generates an increase in physical capacity [38]. However, apparently no studies on the decrease in metabolic oxidation by Spirulina’s administration in elder individuals with excessive body mass have been performed. Hence, well-designed studies are needed to examine the therapeutic value of *S. maxima* supplementation in clinical practice and its synergetic effect with or without exercise to reduce dyslipidemias and body fat in overweight and obese people, a fact almost unknown lately [39]. This establishes that the importance of this study is to contribute in the advancement in the frontiers of knowledge.

### Strengths and Limitations

Even though the present study will show evidence of *S. maxima* supplementation’s clinical importance, the variety of bioactive compounds contained in *S. maxima* limits the possibility of establishing clear and specific action mechanisms. Concerning to the inclusion criteria, only sedentary older adults with BMI over 25 kg/m^2^ will be selected, so the overall results may not translate to other populations of older adults. Another limitation is that the randomization will not be applied to define the inclusion or not in the exercise program intervention, it will be decided by each participant since the personal decision to belong to the exercise group will reduce severely the possibility of study desertion [40,41]. In addition, the beneficial effects associated with healthy lifestyles (physical exercise and nutraceuticals) underlie major causes such as sociodemographic characteristics of the participants, timing of the intervention, dose of supplementation, environmental factors, and genetic background [42]. Previous studies display a considerable amount of risk of bias, such as selection bias (e.g., recruitment of an over-proportionated number of women) and performance bias (e.g., unclear methods of blinding and concealment). Therefore, a proportional number of women and men will be included in this study. Regarding the sample size, the duration of the intervention and the quantity of *S. maxima* to administrate, double-blind studies have reported on the benefits of four weeks of supplementation with nine participants [43], and other studies have reported on the beneficial effects of supplementation with one g/day of *S. maxima* for 12 weeks [44]. All this establishes the importance of this study.

## 4. Conclusions

The SIE study protocol design includes multidisciplinary interventions to define potential future implementation of nutritional supplementation programs with natural products containing a high content of antioxidants and/or proteins, along with personalized exercise programs, in the elderly presenting the main risk factor (excess body fat) of chronic non-communicable diseases. However, possible practical applications could be ensured until results are obtained, where it is expected to have a beneficial impact in all the study’s analysis (body composition, physical capacities, blood lipid profile, mental health, sleep-wake patterns, and inflammatory markers). Finally, the future research lines generated by the SIE project can focus on the inclusion of a more significant number of parameters related to exercise prescription, such as epigenetics, intestinal microbiota, and general fitness. Studies can also be designed to look for long-term effects, including different age groups, or present other pathological conditions. Moreover, the amount of recollected data will allow us to build a dataset, then we will make it available to the Artificial Intelligence community in order to find predictive models.

## Figures and Tables

**Figure 1 ijerph-18-01605-f001:**
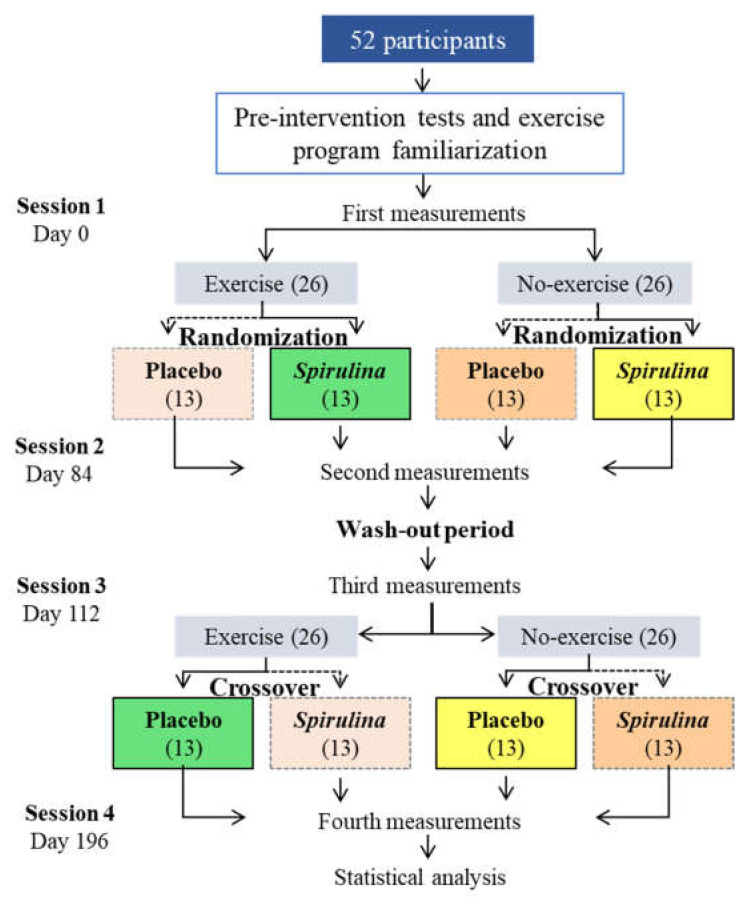
Flow diagram of the experimental design for the impact of *Spirulina maxima* intake and exercise (SIE) trial. Same color indicates the same group of participants.

**Table 1 ijerph-18-01605-t001:** Inclusion and exclusion/elimination criteria to participate in the *Spirulina maxima* intake and exercise (SIE) randomized controlled trial.

Inclusion Criteria	Exclusion/Elimination Criteria
Man or woman	Drinking alcohol
≥60 years	Taking drugs or diet supplements
Daily energy expenditure <4 metabolic equivalents	Presenting an impediment to practicing regular physical exercise
Body mass index≥ 25 kg/m^2^	Participate in any other interventional study
Signed informed consent	<80% attendance to exercise sessions
	Being unhealthy due pre-intervention test results by certified doctor’s decision
	Fracture or surgery in the last six months
	Presenting a chronic disease
	Present a risk when increasing exercise level and/or engage in fitness testing (Physical Activity Readiness-Questionnaire for Everyone (PAR-Q+) with at least one “yes” answer in any of seven pre-screening questions and at least one “yes” in one of the next 10 follow-up questions) [21]

**Table 2 ijerph-18-01605-t002:** Summary of the independent and dependent variables of the *Spirulina maxima* intake and exercise (SIE) randomized controlled trial.

Independent Variables	Dependent Variables
*Spirulina maxima* intake intervention	Body composition (body fat percentage, muscle mass percentage, and body mass index)
Exercise program intervention	Energy intake (total calories, proteins, carbohydrates, fat, and micronutrients consumption)
Wash-out period	Physical function (grip strength and functional physical condition)
Genetic markers	Blood lipid profile (TC, HDL-C, LDL-C, and TG)
	Mental health (anxiety, depression, and cognitive performance)
	Sleep-wake patterns
	Inflammatory markers (IL-6 and TNF-α)

TC: Total cholesterol; HDL-C: High-density lipoproteins associated cholesterol; LDL-C: Low-density lipoproteins associated cholesterol; TG: Triglycerides.

## Data Availability

Any researcher that contacts SIE trial Principal Investigator, M.A.H.-L. (marco.antonio.hernandez.lepe@uabc.edu.mx) will have access to the study data once it has been collected, processed, and analyzed, in accordance with the informed consent provided by participants on the use of confidential data.

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
