# Peer review of "Impact of Spirulina maxima Intake and Exercise (SIE) on Metabolic and Fitness Parameters in Sedentary Older Adults with Excessive Body Mass: Study Protocol of a Randomized Controlled Trial"

_ijerph, 2021, doi:10.3390/ijerph18041605_

Round 1

Reviewer 1 Report

Dear Authors,

I would like to congratulate you for presenting a very interesting future study. I will be looking forward to the results you publish.

However, the protocol needs to be improved:

General comments:

The writing of the article can be improved. Some sentences are too long and some ideas are not adequately expressed.

Introduction:

The introduction is poor in content, there is a lack of epidemiological data and mention of one of the most important conditions among the population under study: sarcopenia.

It is growing evidence that obesity, accompanied by sarcopenia (sarcopenic obesity) is one of the main causes of the disease in the population over 65 years.

Where are the objectives of the work?Even if it is a protocol, the objectives must be set.

Material and Methods:

In ethical aspects, the number and date of approval are missing.

It is not necessary to put the link of the clinical trial in Clinical Trials, only the reference number.

The structure of the section is not adequate. The participants and the inclusion/exclusion criteria should be in section 2.4. In my view, everything related to the characteristics of the clinical trial should appear at the beginning of the material and methods. In sections preceding 2.4, aspects that have not yet been explained are mentioned.

Is the IPAQ validated in Mexico? What is the bibliographical reference?

What is the minimum age for including an elderly person in the study?

What anthropometric variables will be collected in the study? Will they be measured three times and recorded as a mean or only once?

Will the consumption of other sources of antioxidants from the diet be taken into account?

It would be interesting to collect the grip strength, it is a very important variable that is associated with various health disorders and is a good predictor.

What presentation of Spirulina will be used (capsules, powder...)?

The calculation of the sample size is not clearly specified. Please give more details in this section.

Will there be no strength exercise?

I believe that muscle mass should be included as a primary result.

Discusion:

The first paragraph of the discussion should be included in the introduction.

Author Response

We truly appreciate your contributions. The changes requested can be identified in the new version of our manuscript (ijerph-1059026-R1) while a point-by-point response to your requests is described below:

Dear Authors,

I would like to congratulate you for presenting a very interesting future study. I will be looking forward to the results you publish.

R=Thank you.

However, the protocol needs to be improved:

General comments:

The writing of the article can be improved. Some sentences are too long and some ideas are not adequately expressed.

R=According to all your observations, the article has been improved substantially. We really appreciate your contributions.

Introduction:

The introduction is poor in content, there is a lack of epidemiological data and mention of one of the most important conditions among the population under study: sarcopenia.

R= According to your observation, we added information in the introduction focused on sarcopenia, including the following statement: “Some of these conditions can co-exist in aging individuals, presenting new challenges, for example, the combination of excessive body fat and reduced muscle mass or strength, recently defined as sarcopenic obesity. Muscle protein breakdown is one of the main contributors to chronic low-grade inflammation and sarcopenia”.

It is growing evidence that obesity, accompanied by sarcopenia (sarcopenic obesity) is one of the main causes of the disease in the population over 65 years.

R= According to your observation, we have added information in the introduction focused on sarcopenia, including the following statement: “Some of these conditions can co-exist in aging individuals, presenting new challenges, for example, the combination of excessive body fat and reduced muscle mass or strength, recently defined as sarcopenic obesity. Muscle protein breakdown is one of the main contributors to chronic low-grade inflammation and sarcopenia. Recent studies suggest that the progression of muscle fiber anabolism and muscle wasting is highly related to the reduction of the endogen antioxidant system and oxidative stress, inducing the activation of muscular autophagy mechanisms“.

Where are the objectives of the work? Even if it is a protocol, the objectives must be set.
R= According to your comment, the following paragraph has been added: “The main objective of our research is to evaluate the independent and synergistic effect of 12 weeks of S. maxima intake (compared to placebo), with or without a systematic physical exercise program on the body composition (body fat percentage, muscle mass percentage, and body mass index [BMI]) in sedentary older adults with excessive body mass. The secondary objectives are to test the effects of both intervention in the blood lipid profile, physical function, mental health, genetic markers, and inflammation in older adults with excessive body mass

Material and Methods:

In ethical aspects, the number and date of approval are missing.

R= Your observation was considered, so the protocol number and date of approval have been added to the Ethics Section (2.3).

It is not necessary to put the link of the clinical trial in Clinical Trials, only the reference number.

R= The link has already been erased. The reference number has been added, thank you.

The structure of the section is not adequate. The participants and the inclusion/exclusion criteria should be in section 2.4. In my view, everything related to the characteristics of the clinical trial should appear at the beginning of the material and methods. In sections preceding 2.4, aspects that have not yet been explained are mentioned.

R=Thanks for your comment. Your observation was taken into account, so the Clinical Trial Section appear now at the beginning of Materials and Methods.

Is the IPAQ validated in Mexico? What is the bibliographical reference?

R=Yes, it is validated. We added the appropriated reference (Medina, C.; Barquera, S.; Janssen, I. (2013). Validity and reliability of the International Physical Activity Questionnaire among adults in Mexico. Revista Panamericana de Salud Pública, 2013, 34, 21-28)

What is the minimum age for including an elderly person in the study?

R=We truly appreciate your observation, so the participants’ minimum age has already been added in the Participants Section.

What anthropometric variables will be collected in the study? Will they be measured three times and recorded as a mean or only once?

R=Thanks for your comment. Your observation was taken into account, so all the anthropometric procedures have been described, including the following statement: “anthropometry following the standardized protocols of the International Society for the Advancement of Kinanthropometry (ISAK). The anthropometric variables to include are the ones of the restricted profile (height, body mass, skinfolds (tricipital, subscapular, bicipital, suprailiac, supraspinal, abdominal, anterior thigh, and medial calf)), girths (relaxed arm, flexed arm, waist, hip, and calf) and skeletal breadths (humerus and femur). The measurements’ margin of error will be <2% for skinfolds and <1% for girths and skeletal breadths. Anthropometric dimensions will be measured twice (thrice when measurements are greater than the margin of error) and recorded as a mean. Researchers are certified as ISAK anthropometrists and are experienced in handling equipment and procedures”.

Will the consumption of other sources of antioxidants from the diet be taken into account?

R=Yes, they will. They are included in the analysis if micronutrients intake (Section 2.4.2), we really appreciate your observation.

It would be interesting to collect the grip strength, it is a very important variable that is associated with various health disorders and is a good predictor.

R= Your recommendation was considered, so the grip strength has been included in the variables to measure (Section 2.4.3), thank you. (Schaubert, K.L.; Bohannon, R.W. Reliability and validity of three strength measures obtained from community-dwelling elderly persons. J. Strength Cond. Res. 2005, 19, 717).

What presentation of Spirulina will be used (capsules, powder...)?

R= We added the requested information in the Section 2.1, including the following statement: “Presentation of both supplements will be powder encapsulated in dark capsules of 0.5 g to present the same organoleptic attributes and mask their individual characteristics”.

The calculation of the sample size is not clearly specified. Please give more details in this section.

R= According to your comment, information about Sample Size has been updated in Section 2.1.1, including the following statement: “Sample size was calculated considering a level of significance of 0.05 with a study power of 0.85, selecting a sample of >52 participants. The statistical software G*Power [17] was used to determine the sample size”.

Will there be no strength exercise?

R= Yes, there will be strength exercise. We have added the following statement in Section 2.1.3: “a main physical conditioning phase of 25-30 min that will consist of developing a personalized circuit of strength exercises of the main muscle groups (arms, shoulder, chest, back, and legs) at 60% of their 1-Repetition maximum”.

I believe that muscle mass should be included as a primary result.

R= Indeed, it is one of the main outcomes, we have detailed it in Section 2.5, thank you.

Discusion:

The first paragraph of the discussion should be included in the introduction.

R= According to your comment, we changed the paragraph to the Introduction Section.

We appreciate your comments to improve the quality of our manuscript.

Sincerely,

Dr. Luis Mario Gómez-Miranda, Corresponding author

Reviewer 2 Report

The authors present an interesting and novel "study protocol of double-blind, randomized, crossover, controlled trial", interesting the reader to be able to wait for the results of their research. In this sense, I would suggest improving the presentation of antecedents that allow identifying a possible action in adipose tissue of spirulina, alone or in conjunction with physical exercise or in terms of the dose selected for the study.
On the other hand, it would be advisable to incorporate both in the selection of study subjects (because they are elderly), and in the subsequent analysis of the data: the age range, use of medications, multimorbidity, sociodemographic background, etc.

Author Response

We truly appreciate your contributions. The changes requested can be identified in the new version of our manuscript (ijerph-1059026-R1) while a point-by-point response to your requests is described below:

The authors present an interesting and novel "study protocol of double-blind, randomized, crossover, controlled trial", interesting the reader to be able to wait for the results of their research.

R=Thank you.

In this sense, I would suggest improving the presentation of antecedents that allow identifying a possible action in adipose tissue of spirulina, alone or in conjunction with physical exercise or in terms of the dose selected for the study.

R= We considered your suggestion. The Introduction Section has been detailed more clearly, including the following statements: “Some of these conditions can co-exist in aging individuals, presenting new challenges, for example, the combination of excessive body fat and reduced muscle mass or strength, recently defined as sarcopenic obesity. Muscle protein breakdown is one of the main contributors to chronic low-grade inflammation and sarcopenia. Recent studies suggest that the progression of muscle fiber anabolism and muscle wasting is highly related to the reduction of the endogen antioxidant system and oxidative stress, inducing the activation of muscular autophagy mechanisms”; “It has been shown that liver lipid accumulation (as well as the visceral in general), can be reduced by modulating the infiltration of macrophages by Spirulina compounds”. Even, according to the added background, the possibility of carrying out inflammation analysis has been achieved, of which the methodology is described in section 2.4.8.

On the other hand, it would be advisable to incorporate both in the selection of study subjects (because they are elderly), and in the subsequent analysis of the data: the age range, use of medications, multimorbidity, sociodemographic background, etc.

R=Thanks for your comment. Information of the participants’ minimum age and use of medications have been added to Participants Section, and other participants variables will be used like co-variables in the statistical analysis.

We really appreciate your comments to improve the quality of our manuscript.

Sincerely,

Dr. Luis Mario Gómez-Miranda, Corresponding author

Author Response

We truly appreciate your contributions. The changes requested can be identified in the new version of our manuscript (ijerph-1059026-R1), while the 45 comments found in your attached file have a point-by-point response described below:

Comments 1, 5, 14, and 35: recommendations of changing “weight” for “body mass”

R=Thanks for your comments. Your observation was considered, so we have changed the requested sentences.

Commentary 2: Include the main goal in line 22

R= According to your comment, the following paragraph has been added: “The main objective of our research is to evaluate the independent and synergistic effect of 12 weeks of S. maxima intake (compared to placebo), with or without a systematic physical exercise program on the body composition (body fat percentage, muscle mass percentage, and body mass index [BMI]) in sedentary older adults with excessive body mass. The secondary objectives are to test the effects of both intervention in the blood lipid profile, physical function, mental health, genetic markers, and inflammation in older adults with excessive body mass”.

Commentary 3 and 4: Correction of the affirmation “will be evaluated” in line 25 to past sentence and include numeric numbers abut the main results in line 26

R= This study has not been conducted yet, it is only a study protocol. The World Health Organization and the International Committee of Medical Journal Editors highly recommend and support to register and publish clinical trial protocols to improve study planning, conduction, reporting and appraisal (Summerskill, W.; Collingridge, D.; Frankish, H. Protocols, probity, and publication. The Lancet 2009, 373, 992).

Commentary 6: Delete sentences no appropriated in abstract section

R= The required action has been taken. Thanks for your observation.

Commentary 7: Reconsider  keyword “Randomized clinical trial”

R= Your recommendation was considered. We have changed the keyword “Randomized clinical trial” for “Randomized trial”, thank you.

Commentary 8: Use capita letter in “world”

R= The required action has been taken. Thanks for your observation.

Commentary 9: Better “population in the next years” than “population”

R= The required action has been taken. Thanks for your observation.

Commentary 10: Include reference in line 40

R= The requested reference has been added. Thank you (Rodgers, J.L.; Jones, J.; Bolleddu, S.I.; Vanthenapalli, S.; Rodgers, L.E.; Shah, K.; Karia, K.; Panguluri, S.K. Cardiovascular risks associated with gender and aging. J. Cardiovasc. Dev. Dis. 2019, 6, 19).

Commentary 11: Change the colloquial sentence “this and others”

R= According to your observation, the colloquial sentence was changed. Thank you so much.

Commentary 12: “Given that” better than “Because”

R= Your observation was considered and the phrase has been changed. Thank you.

Commentary 13: What is “ta” in line 56

R= We had a mistake in that word; the correct term was “it,” of any form; we have changed the sentence to the following: “Furthermore, performing systematic physical exercise has been widely reported to reduce the risk of hypertension, diabetes, and hypercholesterolemia”, thanks for your observation.

Commentary 15: Section 2.2 (Participants) better before Section 2.1 (Ethics)

R= According to your observation, we have reorganized those two Sections. Thank you.

Commentary 16: Include ethics reference of the World Medical Association (Declaration of Helsinki, 7th edition).

R= The requested reference was added in Ethics Section. Thank you (Hellmann F.; Verdi M.; Schlemper B.R.Jr; Caponi S. 50th anniversary of the Declaration of Helsinki: the double standard was introduced. Arch Med Res. 2014, 45, 600-601).

Commentary 17: Include main characteristics of participants

R= The requested information was added in Participants Section (minimum age and body mass index). Thank you.

Comments 18 and 29: “Participants” better than “subjects”

R= Your observation was considered and that word has been changed throughout the document. Thank you.

Commentary 19: Include a general procedure section in line 90

R= The requested information was added in Section 2.4, including the following paragraph: “Eligible participants will visit the body composition laboratory at UABC in Tijuana, Mexico, between 7-9 AM for baseline measurements, one the day before the onset of the treatment. Standard room temperature and humidity will be checked constantly. First the antecubital blood samples will be drawn by an expert clinician into EDTA tubes after overnight fasting, after that, body composition measures will be performed, finally, dietary, psychological, and physical activity questionaries will be obtained. All procedures will be repeated on days 84, 112, and 196”

Commentary 20: Include reference of the ISAK protocol in line 102

R= The requested reference was added in Section 2.4.1. Thank you (Ross, W.D.; Marfell-Jones, M.J. Kinanthropometry. In Physiological Testing of Elite Athlete; Human Kinetics Publishers Inc.: London, UK, 1991).

Commentary 21: Include number registration of the ISAK level two Anthropometrist

R= Actually, ISAK does not give number registration for their members, but according to your requirement, we attached the Anthropometrists certificates in the Supplementary File 4. Also, you can find the information on the official page of ISAK (https://www.isak.global/MemberList/).

Commentary 22: Explain the variables in line 100

R= The requested information was added in Participants Section (minimum age and body mass index). Thanks for your observation.

Comments 23, 27, and 28: Correct the time in the sentence “will be applied”

R= This study has not been conducted yet, it is only a study protocol. The World Health Organization and the International Committee of Medical Journal Editors highly recommend and support to register and publish clinical trial protocols to improve study planning, conduction, reporting and appraisal (Summerskill, W.; Collingridge, D.; Frankish, H. Protocols, probity, and publication. The Lancet 2009, 373, 992).

Commentary 24: Include reference in dietary analysis

R= The requested reference was added in Section 2.4.2. Thank you (Nelson, M. The validation of dietary assessment. In Design Concepts in Nutritional Epidemiology, 2nd ed.; Margetts, B.M., Nelson, M., Eds.; Oxford University Press: Oxford, UK, 1997; 266–295).

Commentary 25: “physical capacities” better than “physical function”

R= Your observation has been considered and we have renamed Section 2.4.3. Thank you.

Commentary 26: Include validity and reliability of each test with reference

R= The requested references about validity and reliability were added in Section 2.4.3. Thank you.

Schaubert, K.L.; Bohannon, R.W. Reliability and validity of three strength measures obtained from community-dwelling elderly persons. J. Strength Cond. Res. 2005, 19, 717.

Rikli, R.E.; Jones, C.J. Senior Fitness Test Manual. Human Kinetics. Champaign, IL, USA. 2013.

Commentary 30: Include validity of sleep device

R= The requested reference about validity of the sleep device has been added in Section 2.4.6. Thank you (Lichstein, K.L.; Stone, K.C.; Donaldson, J.; Nau, S.D.; Soeffing, J.P.; Murray, D.; Lester, K.W.; Aguillard, R.N. Actigraphy validation with insomnia. Sleep 2006, 29, 232-239).

Commentary 31: Delete one coma in line 156

R= Many thanks for your observation, it was done.

Commentary 32: Describe which the criteria to select these genes

R= Thank you for your comment. The FTO and FABP2 gene polymorphisms were selected based on previous candidate-gene studies showing their association with obesity traits and blood lipid profile in Mexican population, respectively.

References

Gonzalez-Becerra, K.; Ramos-Lopez, O.; Garcia-Cazarin, M.L.; Barron-Cabrera, E.; Panduro, A.; Martinez-Lopez, E. Associations of the lipid genetic variants Thr54 ( FABP2) and -493T ( MTTP) with total cholesterol and low-density lipoprotein cholesterol levels in Mexican subjects. J. Int. Med. Res. 2018, 46, 1467-1476.

Martínez-López, E.; Ruíz-Madrigal, B.; Hernández-Cañaveral, I.; Panduro, A. Association of the T54 allele of the FABP2 gene with cardiovascular risk factors in obese Mexican subjects. Diab. Vasc. Dis. Res. 2007, 4, 235-6.

Villalobos-Comparán, M.; Flores-Dorantes M.T.; Villarreal-Molina M.T.; Rodríguez-Cruz, M.; García-Ulloa, A.C.; Robles, L.; Huertas-Vázquez, A.; Saucedo-Villarreal, N.; López-Alarcón, M.; Sánchez-Muñoz, F.; Domínguez-López, A.; Gutiérrez-Aguilar, R.; Menjivar, M.; Coral-Vázquez, R.; Hernández-Stengele, G.; Vital-Reyes, V.S.; Acuña-Alonzo, V.; Romero-Hidalgo, S.; Ruiz-Gómez, D.G.; Riaño-Barros, D.; Herrera, M.F.; Gómez-Pérez, F.J.; Froguel, P.; García-García, E.; Tusié-Luna, M.T.; Aguilar-Salinas, C.A.; Canizales-Quinteros, S. The FTO gene is associated with adulthood obesity in the Mexican population. Obesity 2008, 16, 2296-301.

Commentary 33: Why 12 weeks?

R= Times have been defined about a previously published systematic review (Hernández-Lepe, M.A.; Wall-Medrano, A.; Juárez-Oropeza, M.A.; Ramos-Jiménez, A.; Hernández-Torres, R.P. Spirulina and its hypolipidemic and antioxidant effects in humans: a systematic review. Nutr. Hosp. 2015, 32, 494-500), we have added the following statement in Section 2.1. “Due to absence of data related to washout and treatment periods length, we considered 4 and 12 weeks, respectively, as stated by a systematic review of clinical trials that used spirulina as treatment”.

Commentary 34: Describe the warm up

R= The requested information was added in Section 2.1.3, including the following statement: “warm-up phase of 5-10 min (joint movement exercises to increase body temperature starting with movements of head, upper extremities trunk, hip, and, lower extremities)”

Commentary 36: Include when non parametric variables

R= The requested information has been added in Section 2.6, including the following statement: “Kruskal-Wallis H and Dunn’s Post-hoc test (non-parametric distribution and unequal variances), with Bonferroni setting for multiple comparisons “. Thank you.

Commentary 37: Delete one point in line 238

R= Many thanks for your observation.

Commentary 38: Include the sentence “All statistical analyses were completed using IBM SPSS package version 23 (IBM Corporation, 186 Chicago, IL, USA)” at the end of the paragraph in line 240

R= The requested sentence has been added in Section 2.6. Thanks for your contribution.

Commentary 39: Include a p value in line 240

R= The requested information has been added in Section 2.6 in the following statement: “Statistically significance will be set at a p-value of 0.05”.

Commentary 40: Where are the results section?

R= This is only a study protocol, the study will be conducted in January 2022, so we really regret not adding results information yet.

Commentary 41: Where are the Tables and Graphs

R= This is just a study protocol; the study will be performed in January 2022, so we really regret not adding results tables and graphs yet; We only present the protocol flow chart in Figure 1.

Commentary 42: Discussion section start with the main aims and main results

R= According to your comment, we have changed the Discussion first paragraph to the Introduction Section. Thank you

Commentary 43: Where are the summary section?

R= According to your request, we have added a Conclusion Section, including the following: “The SISPE study protocol design includes multidisciplinary interventions to define potential future implementation of nutritional supplementation programs with natural products containing a high content of antioxidants and/or proteins, along with personalized systematic exercise programs, in the elderly presenting the main risk factor (excess body fat) of chronic non-communicable diseases. However, possible practical applications could be ensured until results are obtained, where it is expected to have a beneficial impact in all the study's analysis (body composition, physical capacities, blood lipid profile, mental health, sleep-wake patterns, and inflammatory markers). Finally, the future research lines generated by the SISPE project can focus on the inclusion of a more significant number of parameters related to exercise prescription, such as epigenetics, intestinal microbiota, and general fitness. Studies can also be designed to look for long-term effects, including different age groups, or present other pathological conditions. Moreover, the amount of recollected data will allow us to build a dataset, then we will make it available to the Artificial Intelligence community in order to find predictive models.

Commentary 44: Future research lines

R= Your request was added in the Conclusion section, including the following statement: “Finally, the future research lines generated by the SISPE project can focus on the inclusion of a more significant number of parameters related to exercise prescription, such as epigenetics, intestinal microbiota, and general fitness. Studies can also be designed to look for long-term effects, including different age groups, or present other pathological conditions. Moreover, the amount of recollected data will allow us to build a dataset, then we will make it available to the Artificial Intelligence community in order to find predictive models”.

Commentary 45: Include practical application paragraph

R= The requested information was added in Conclusion Section, including the following statement: “The SISPE study protocol design includes multidisciplinary interventions to define potential future implementation of nutritional supplementation programs with natural products containing a high content of antioxidants and/or proteins, along with personalized systematic exercise programs, in the elderly presenting the main risk factor (excess body fat) of chronic non-communicable diseases. However, possible practical applications could be ensured until results are obtained, where it is expected to have a beneficial impact in all the study's analysis (body composition, physical capacities, blood lipid profile, mental health, sleep-wake patterns, and inflammatory markers)”.

We really appreciate your comments to improve the quality of our manuscript.

Sincerely,

Dr. Luis Mario Gómez-Miranda, Corresponding author

Round 2

Reviewer 3 Report

Accepted for publication

Author Response

We really appreciate your contributions to improve the quality of our manuscript.

This manuscript is a resubmission of an earlier submission. The following is a list of the peer review reports and author responses from that submission.